# Impact of Host, Lifestyle and Environmental Factors in the Pathogenesis of MPN

**DOI:** 10.3390/cancers12082038

**Published:** 2020-07-24

**Authors:** Gajalakshmi Ramanathan, Brianna M Hoover, Angela G Fleischman

**Affiliations:** 1Division of Hematology/Oncology, Department of Medicine, University of California, Irvine, CA 92617, USA; agf@uci.edu; 2Department of Biological Chemistry, University of California, Irvine, CA 92617, USA; bcraver@uci.edu

**Keywords:** myeloproliferative neoplasm, lifestyle, smoking, inflammatory diet, prevention

## Abstract

Philadelphia-negative myeloproliferative neoplasms (MPNs) occur when there is over-production of myeloid cells stemming from hematopoietic stem cells with constitutive activation of JAK/STAT signaling, with *JAK2^V617F^* being the most commonly occurring somatic driver mutation. Chronic inflammation is a hallmark feature of MPNs and it is now evident that inflammation is not only a symptom of MPN but can also provoke development and precipitate progression of disease. Herein we have considered major MPN driver mutation independent host, lifestyle, and environmental factors in the pathogenesis of MPN based upon epidemiological and experimental data. In addition to the traditional risk factors such as advanced age, there is evidence to indicate that inflammatory stimuli such as smoking can promote and drive MPN clone emergence and expansion. Diet induced inflammation could also play a role in MPN clonal expansion. Recognition of factors associated with MPN development support lifestyle modifications as an emerging therapeutic tool to restrain inflammation and diminish MPN progression.

## 1. Introduction

Myeloproliferative neoplasms (MPNs) occur as a result of clonal outgrowth of hematopoietic stem cells (HSCs) carrying a somatic mutation, most commonly in *JAK2* (*JAK2^V617F^*), endowing them with cytokine independent growth and ensuing in the overproduction of mature and differentiated cells of cells of the myeloid lineage [1,2,3,4,5]. MPNs are distinct from myelodysplastic syndromes (MDS) and acute myeloid leukemia (AML) which are characterized by arrested differentiation of progenitor cells leading to bone marrow failure and ineffective hematopoiesis. Myeloproliferative neoplasms (MPNs) are classic examples of chronic inflammation driven cancers where the inflammatory environment impacts disease initiation, symptomatology, and progression. 

Although there is a germline predisposition to acquire MPNs with familial clustering [6,7], many MPN cases are sporadic and occur in aging adults. Chronic inflammation is a consequence of both intrinsic and extrinsic factors. Defining how risk factors impact MPN development will enable lifestyle modifications to improve patient outcomes in early stage of disease as well as to prevent MPN initiation in healthy individuals predisposed to developing MPN. In this review we will focus on the host components, lifestyle factors and environmental drivers of inflammation and their associated risk in MPN pathogenesis (Figure 1).

## 2. Host Factors

### 2.1. Clonal Hematopoiesis of Indeterminate Potential (CHIP)

Clonal hematopoiesis (CH) is the expansion of peripheral blood cells derived from a single HSC due to the acquisition of a somatic mutation that improves HSC fitness relative to adjacent stem cells and contributes to a selective growth advantage [8]. Clonal hematopoiesis of indeterminate potential (CHIP) is defined as the presence of at a myeloid malignancy associated gene mutation with a variant allele frequency (VAF) of at least 2% [8,9]. This time-dependent acquisition of somatic mutations in HSCs establishes aging as the strongest predictor of CHIP where it is rare under the age of 40 but more prevalent, at a frequency of at least 10%, in individuals over the age of 70 [8,10]. The presence of CHIP significantly increases the risk for developing a hematologic malignancy but this outcome only occurs in a fraction of those with CHIP [8,11]. 

Somatic mutations in epigenetic regulatory genes such as *DNMT3A* and *TET2* along with *JAK2^V617F^* are among the most common CHIP mutations [8]. The presence of CHIP mutations is associated with inflammatory markers such as increased serum IL-6, TNF-α, IL-8 and high-sensitivity C-reactive protein (hsCRP) [12,13,14]. Compared to the incidence of CHIP in the general population from prior studies and published cohorts, the frequency of CHIP was higher in chronic inflammatory conditions such as aplastic anemia [15,16], ulcerative colitis patients [17] and in patients with aortic valve stenosis [18].

However, whether CHIP-associated inflammation drives progression to hematologic malignancy has not been established. Mechanistic studies in *TET2* knockout mice indicate that the relationship between CHIP and inflammation is bidirectional. TET2 function is required to restrain inflammatory gene expression of IL-6 and IL-1β in macrophages [19], resolve inflammation and repress IL-6 expression in myeloid cells [20] and regulate the NLRP3 inflammasome-mediated IL-1β secretion [21,22] but *TET2* deficient mice also show expansion of hematopoietic stem cells (HSCs) and differentiated myeloid cells in response to inflammatory stress in combination with an enhanced production of pro-inflammatory IL-6 [23]. It can thus be recognized that CHIP mutations by themselves can induce a low-grade inflammatory state but the presence of additional stressors like infection or autoimmune disease may exaggerate CHIP induced inflammation, thus augmenting the selective advantage for the mutant clone. 

Blood cells carrying CHIP mutations survive well in a pro-inflammatory environment, further reiterating that an inflammatory milieu strengthens the selection advantage of CHIP clones. *JAK2*-mutated progenitor cells from MPN patients have increased myeloid colony formation in the presence TNF-α compared to unmutated *JAK2* cells from these same patients and normal controls [24]. Similarly, *TET2* mutant bone marrow cells from myelodysplastic syndrome (MDS) subjects are resistant to the suppressive effects of TNF-α on colony formation [25]. Thus, the survival and/or proliferation of the *JAK2^V617F^* and *TET2* mutated clones are driven by inflammation. It is possible that specific types of inflammation drive the selective advantage of specific mutant clones. 

### 2.2. Infections and Auto-Immune Disorders

Chronic immune stimulation through infection or autoimmune disease may act as a trigger for myeloid malignancy development [26]. In a large Swedish population-based study a history of infection as well as autoimmune disease were both associated with an increased risk of Acute Myeloid Leukemia (AML) and MDS [26]. Using the SEER-Medicare database Anderson et al. found that autoimmune conditions were associated with an increased risk of acute myeloid leukemia (AML) and myelodysplastic syndrome (MDS) [27]. Kristinsson et al. found a prior history of any autoimmune disease to be associated with a significantly increased risk of MPN [28]. Several infections, particularly those affecting the respiratory tract, were associated with AML and MDS even when they occur many years before diagnosis [29]. However, MPN was only significantly associated with a history of cellulitis [29]. These studies suggest that different types of infection may create a specific inflammatory milieu selective for specific mutant driver clones.

Autoimmune disease, most notably inflammatory bowel disease, has been associated with an increased risk of clonal hematopoiesis. Analysis of patients with thrombocytosis in an inflammatory bowel clinic revealed that 23% of them harbored *JAK2^V617F^* mutations [30], this highlights that some cases of assumed reactive thrombocytosis in a patient with an inflammatory disease may actually be MPN. Also, a cohort of patients with ulcerative colitis was found to have an increased rate of CHIP, in particular with *DNMT3A* and *PPM1D* mutations [17]. Moreover, there was a specific association between elevated levels of serum interferon gamma and *DNMT3A* mutations. 

### 2.3. Aging

The gain-of-function *JAK2^V617F^* mutation is the most common acquired somatic mutation in MPN. Age is a well-defined risk factor for *JAK2^V617F^*-positive MPN typically presenting at older age [31,32]. The MPN Research Initiative sponsored by 23andMe showed strong age interaction in the epidemiology of MPN with 41.1% of MPN patients being between 61–112 years of age compared to 30.5% of the population controls being over the age of 60 [33]. The *JAK2^V617F^* mutation can also be detected in the general population. The Copenhagen General Population Study and 23andMe identified *JAK2^V617F^* carriers with a prevalence of 0.17% to 0.20% in the general population [33,34]. The prevalence of *JAK2^V617F^* was associated with increasing age and male sex in both studies [33,34]. The Danish General Suburban Population Study screened almost 20,000 citizens for *JAK2^V617F^* and *CALR* mutations using droplet digital polymerase chain reaction and found 3.2% and 0.16% of this population to harbor *JAK2^V617F^* and *CALR* mutant cells, respectively [35]. Increasing age, smoking, and alcohol were risk factors for the mutations in this study.

### 2.4. Microbiome

The human body contains over 40 trillion microbes with a vast majority residing in the gastrointestinal tract, referred to as gut microbiota, and colonization with a large and diverse commensal microbiota population is beneficial to health. The human intestinal gut microbiome is composed mainly of bacteria and has an important influence in many physiological functions such as metabolism, inflammation, and hematopoiesis. In mice, the gut microbiome directs innate immune cell development by promoting steady-state myeloid cell development in the bone marrow [36]. Germ-free mice also display defects in adaptive immune cell populations indicating the importance of the microbiome in regulating hematopoiesis [37]. Furthermore, antibiotic induced depletion of the intestinal microbiota not only leads to multilineage suppression of bone marrow hematopoiesis under steady-state [38] but also impairs hematopoietic recovery following bone marrow transplant in a lethally irradiated murine model [39].

Defining the microbiome’s role in the pathogenesis of hematologic malignancies could lead to improved diagnostic, treatment, and prevention options. A global reduction in the abundance and diversity of intestinal microbiota is characteristic in Crohn’s disease [40] which in turn is associated with increased risk for MPN [28]. Thus, dysbiosis of the gut microbiome could be a trigger for MPN disease progression. To date, no large exploratory studies have been undertaken to investigate changes in the microbiome composition between MPN and normal subjects. In a pilot study comparing the gut microbiome of MPN versus healthy controls, we found *Phascolarctobacterium* to be more abundant in healthy over MPN (Fleischman lab, unpublished data). *Phascolarctobacterium* are underrepresented in inflammatory diseases such as primary sclerosing cholangitis and ulcerative colitis [41] and necrotizing enterocolitis [42] suggesting that *Phascoloarctobacterium* may be beneficial to reduce inflammation. Further studies with larger cohorts are required to address the causal role of microbiota in driving inflammation in MPN. In this regard, a recent study by Meisel et al demonstrated that microbial signals from the gut are necessary to drive pre-leukemic myeloproliferation in *TET2* knockout mice [43]. This study showed that microbial translocation caused by a loss of integrity in the intestinal barrier results in IL-6 induced inflammation and pre-leukemic transformation. Interestingly, this phenomenon was not observed in germ-free mice or mice treated with antibiotics, indicating that the microbiome may play an instructive role in the development of myeloid malignancies [43]. Another study also showed that antibiotic treatment could suppress the development of myeloid malignancy in *TET2* knockout mice by modulating the TNF-α signaling pathway [44]. Though steady-state and stress hematopoiesis benefit from a diverse and abundant microbiome, modulation of the intestinal flora could possibly be used to prevent leukemic transformation in the presence of myeloid malignancy related candidate driver genes. Moreover, intestinal microbial communities could also potentially be targeted to manage or prevent chronic inflammatory conditions such as MPN. These approaches include fecal microbial transplantation [45], antibiotics [44] and dietary interventions [46].

## 3. Lifestyle Factors

### 3.1. Smoking

Cigarette smoking causes considerable morbidity and mortality by promoting development of chronic conditions such as cardiovascular diseases and cancer. Smoking is known to increase the risk for developing acute myeloid leukemia (AML) in adults [47,48]. Recently, a growing number of reports indicate a predisposition towards increased risk for developing MPN in ever-smokers. A recent report demonstrates that current and heavy smokers in the general population are more likely to carry a *JAK2^V617F^* or *CALR* mutation [35]. Smoking also promotes systemic chronic inflammation via leukocyte activation, endothelial dysfunction, oxidative stress and release of pro-inflammatory cytokines, all characteristic features of MPN [49]. These support the idea that smoking induces mutagenesis in HSCs leading to the creation of a mutant clone and/or produces an environment which allows mutant clones to gain a selective advantage. Coombs et al. observed an increased rate of smoking associated mutational signature in CHIP mutations among smokers with solid tumors [50]. Meanwhile, although ASXL1 mutations were predominantly associated with smoking in the UK Biobank cohort, the rate of C > A mutations, characteristic of the smoking related mutational signature 4, was similar between smokers and non-smokers [51]. In addition, mutational signature 4 was mainly found in tissues directly exposed to tobacco smoke and could not be extracted from malignancies such as acute myeloid leukemia, despite the known risk associated with smoking [52]. Thus, it appears that a smoking induced inflammatory environment allows clonal evolution and MPN disease progression. 

The effects of smoking also include changes in peripheral blood cell counts and hematological parameters such as erythrocytosis and leukocytosis [53,54]. Pasqualetti et al., reported a positive association between heavy smokers and hematologic malignancy [55], particularly MDS. The UK Million Women Study found a positive correlation between exposure to tobacco smoke and the incidence of MPN. This large prospective study reported that current smokers had a higher risk for developing MDS and MPN compared to never-smokers [56]. Women smoking <15 cigarettes per day had a relative risk of 1.52, (95% Confidence Interval (CI) 1.27–1.81) and almost double the risk was observed in frequent smokers (>15 cigarettes/day) Relative Risk (RR) = 1.98 (95% CI 1.67–2.35) using never-smokers as a reference group [56]. In the Iowa Women’s Health Study, current smokers presented with an increased risk for all MPNs, RR = 1.72 for current versus never smokers. For the particular subtypes, the association of current smoking was stronger for Polycythemia Vera (PV) incidence than Essential Thrombocythemia (ET) [57]. 

Using chronic lymphoid leukemia (CLL) patients as controls, Sørensen and Hasselbalch conducted a case-control study to determine the relationship between smoking and MPN. When former and current smokers were compared to never-smokers, there was a significant association between smoking history and the risk of MPN compared to CLL controls (Overall Risk (OR) = 1.73, 95% CI 1.25–2.40) [58]. The association was significant for both ET and PV subtypes. Data from the Danish Health Examination Survey, a general population-based study, was also used to establish whether smokers had an increased risk for MPN compared to non-smokers. The risk for MPN in daily smokers was more than doubled, with a hazard ratio 2.5 (95% CI 1.3–5.0) and 1.9 (95% CI 1.1–3.3) for occasional/ex-smokers. The risk for smoking associated MPN was dose-dependent, an increased amount of daily smoking increased the hazard ratio for any MPN compared to non-smokers [59]. A recent meta-analysis that combined several published studies also reported an increased odds ratio for MPN when comparing smokers with non-smokers [54]. The odds ratio for MPNs were increased when comparing any type of smoking history; current, former, light and heavy smoker with non-smoker. Heavy smokers had double the risk for MPN compared to never-smokers [54]. 

Two very recent studies also found smoking to be a modifiable risk factor for MPN development. The NIH-AARP Diet and Health Study identified that smoking behavior increased the risk for MPN in women [60]. This study showed that both former and current female smokers had an increased risk for developing MPN compared to never-smokers [60]. The MOSAICC (MyelOproliferative neoplasmS: An In-depth Case-Control) study also confirmed that current smoking status significantly associated with PV (OR = 3.73, 95% CI 1.06–13.15) [61]. Also, heavy smokers described as smoking 20 or more pack years of cigarettes were associated with increased MPN risk in this study [61]. 

While it is well established that conventional combustible smoking causes significant negative health outcomes, the impact of electronic cigarettes (E-cigs) is unknown. To investigate the impact of E-cigs on normal hematopoiesis, we exposed wild-type mice to two months of E-cig smoke and found significant decreases in the bone marrow populations of myeloid progenitors, specifically the common myeloid progenitors. Furthermore, when E-cig exposure was followed by an acute inflammatory challenge, transplanted HSCs showed inferior engraftment at early time points and delayed monocytosis (Ramanathan et al., submitted). These findings indicate that chronic E-cig use may affect hematopoietic stem and progenitor cells and need to be used with caution. 

### 3.2. Diet

The NIH-AARP Diet and Health Study examined the influence of dietary factors in association with risk for chronic myeloid leukemia (CML) and found that none of the wide range of dietary factors studied were associated with risk for CML [62]. Recently, data from the NIH-AARP study was used to examine lifestyle factors and MPN risk. Coffee consumption was found to be inversely related to MPN risk where compared to low level of caffeine intake, high intake of caffeine was protective against risk for PV development [60]. 

Inflammation can be modulated through diet, and so dietary intervention may represent a low-risk way to manage inflammation in MPN. A Mediterranean dietary pattern scientifically helps protect against major chronic diseases like obesity, type-2 diabetes, and inflammatory disorders [63]. The PREDIMED (Prevención con Dieta Mediterránea) trial assessed the Mediterranean diet’s anti-inflammatory properties among participants who were at high cardiovascular risk but did not have known cardiovascular disease. They were randomized into two different Mediterranean diet arms (one arm supplemented with extra-virgin olive oil; the other supplemented with mixed nuts) and one control arm. Patients in both Mediterranean diet groups showed a reduced incidence of major cardiovascular events (hazard ratio 0.70 for group assigned to Mediterranean diet with extra-virgin olive oil and 0.72 to the group assigned to Mediterranean diet with nuts) [64]. Participants who were adherent to the Mediterranean diet significantly decreased their levels of CRP, IL-6 and endothelial and monocyte adhesion molecules and chemokines [65].

To determine if MPN patients can adopt a Mediterranean style eating pattern if given dietician counseling and written curriculum we developed a prospective interventional proof-of-concept 15-week study of 28 MPN patients randomized (1:1) at enrollment to either a Mediterranean (MED) diet supplemented with Extra Virgin Olive Oil (EVOO) or the United States Dietary Guidelines for Americans (USDA). Conformity with the Mediterranean dietary pattern was assessed by the 14-item Mediterranean diet adherence score (MEDAS) [66], with a MEDAS score of ≥8 defined as adherence to a Mediterranean diet. Throughout the entire intervention phase at least 75% of patients in the Mediterranean diet arm achieved a MEDAS score of ≥8. In contrast, during the intervention phase 15–46% of the USDA arm achieved a score of ≥8 (unpublished data, Fleischman lab). In this study we also explored the impact of diet on symptom burden. Out of all the participants that scored ≥8 in the MEDAS, 42% had a ≥50% decrease in symptom burden at week 9 (during the dietary intervention) compared to baseline (weeks 1 and 2) regardless of randomization. These observations suggest that a high Mediterranean diet adherence score may correlate with a lower symptom burden. 

Specific dietary components may promote healthy hematopoiesis. The amino acids valine and cysteine are essential for the maintenance of hematopoiesis [67]. Hematopoietic stem and progenitor cells showed significant growth and proliferative retardation when cultured in medium lacking cysteine while valine effected stem but not progenitor cells. Similarly, in competitive bone marrow transplants, mice that received HSCs cultured in the absence of valine or cysteine showed no engraftment demonstrating that these amino acids are required for HSC maintenance. Interestingly, the anti-oxidant, *N*-acetylcysteine (N-AC), rescued HSC growth in cysteine lacking medium [67]. We have also shown that N-AC is particularly useful in mitigating thrombosis in a mouse model of MPN [68] and further investigation of N-AC in promoting stem cell fitness in the setting of MPN is required. 

Vitamins A, C and D have been implicated in hematopoiesis [69,70,71]. Vitamin C or ascorbic acid is a co-factor of TET2 enzymatic activity and limits HSC self-renewal and suppresses leukemogenesis [70]. Vitamin C supplementation blocked myeloid disease development in mice and controlled disease progression in AML xenograft models [72]. A clinical trial in elderly AML patients demonstrated that the addition of intravenous vitamin C to decitabine resulted in a higher complete remission (CR) than decitabine alone (79.92% vs. 44.11%; *p* = 0.004) after one cycle of chemotherapy [73]. A low vitamin D diet was shown to prevent myelofibrosis in a *JAK2^V617F^* transgenic mouse model [74] although MPN patients do not show elevated serum vitamin D levels [75]. These reports indicate that it is possible that nutritional status can influence MPN disease development and progression. 

Other dietary compounds that have been reported to have effects on hematopoiesis include resveratrol and curcumin. Resveratrol is a naturally occurring plant-derived polyphenol and resveratrol treatment increased bone marrow HSPCs [76] as well as protected HSCs from radiation induced injury [77]. Curcumin treatment in vitro induced apoptosis in leukocytes from *JAK2^V617F^* positive MPN patients [78]. These compounds are known for their anti-inflammatory properties and could have potentially beneficial effects in lowering the risk for MPN. More research on the metabolic requirements and vulnerabilities of normal and *JAK2^V617F^* mutated HSCs will provide new avenues of modulating MPN disease development. 

### 3.3. Obesity and Physical Activity

The UK Million Women Study was a prospective study with a 10-year follow-up period. This study showed an increased relative risk (RR) of MPN or MDS (RR 1.32, 95% CI 1.15–1.52) for an increase in 10 kg/m^2^ body-mass index (BMI) [79]. Similarly, the Iowa Women’s Health Study, also a prospective study cohort with follow up period of 11 years, found that the relative risk for ET was positively associated with baseline BMI, increasing risk for ET with increasing BMI [57]. Interestingly, neither BMI nor body weight at study baseline were associated with risk for PV development [57]. More recently, the MOSAICC case-control study also reported that obesity appeared to elevate overall risk for MPN, and among the sub-types obesity was significantly associated with ET alone (OR 2.59, 95% CI 1.02–6.58) [61]. High intensity physical activity was in turn inversely correlated with ET development (RR 0.66, 95% CI 0.44–0.98) but not PV in the Iowa Women’s Health Study [57]. The disparity between the risk for ET versus PV due to obesity is interesting. One possible mechanism could be due to the influence of increased adiposity in the bone marrow which leads to enhanced megakaryocyte maturation and accelerated proplatelet formation in mice [80]. Interventions using yoga in MPN have been performed [81,82], however no interventional studies using more rigorous physical activity with the intent of weight reduction in MPN have been performed. 

Recent animal studies have shown that diet-induced obesity negatively affects HSC activity. For instance, a diet high in animal fat (i.e., lard) promoted expansion of pro-inflammatory myeloid cell production from the bone marrow [83]. Spred1 (Sprouty-related, EVH1 domain-containing 1) regulates RAS-MAPK signaling and under steady-state negatively influences HSC self-renewal. A lard based high fat diet induces systemic stress in Spred1 deficient mice leading to increased leukocytes and thrombocythemia in peripheral blood along with marked splenomegaly. Specifically, a fraction of Spred1 deficient mice fed an animal fat diet develop an ET-like phenotype without myeloid cell abnormalities [84]. This study also demonstrated that depletion of microbiota in Spred1 deficient mice fed a lard fat-based diet rescued the MPN-like disease phenotype implicating gut microbiota dysbiosis in triggering MPN-like disease [84].

Mice fed an inflammatory high fat diet also displayed poor hematopoietic recovery following a hematologic stress challenge [85]. Obesity related alterations in the gut microbiome have been attributed to poor HSC function by increasing BM-adipocyte differentiation [85]. Thus, changes in the bone marrow microenvironment caused by an inflammatory diet affect HSC self-renewal and an anti-inflammatory or low inflammatory index diet such as the Mediterranean diet could be a possible strategy to prevent HSC anomalies. 

## 4. Environmental Factors

### 4.1. Ionizing Radiation

Acute exposure to high doses of atomic bomb radiation among Japanese residents is associated with AML and CML [86] while protracted low-dose radiation exposure also increases the risk for CML [87]. Although the incidence of MPN following low-dose radiation is not reported, the genomic characteristics of MPN patients with and without exposure to radiation from the Chernobyl nuclear accident have been evaluated. The study found that ionizing radiation exposed MPN patients display a different genetic profile compared to unexposed patients where the incidence of *JAK2^V617F^* mutation was significantly less in the radiation exposed patients. Other findings included a higher rate of type-1 like *CALR* mutations and triple negative MPN in the ionizing radiation exposed group versus unexposed patients [88]. A study by Mele et al., reported that living in tuff houses, a building material containing gamma-emitting radionuclides, increases the risk for ET [89]. Meanwhile, Najean et al., reported no association between radiation exposure and PV [90]. The MOSAICC study also reported that MPN patients were likely to have had three or more CT scans compared to controls with odds ratio of 5.38 (95% CI 1.67–17.3) [61]. Further in-depth evaluation of low-dose radiation exposure on MPN development is required to determine the true existence of an increased risk and causal relationship. 

### 4.2. Occupational and Chemical Exposure

The effect of benzene exposure on developing MPNs has been investigated in terms of occupational exposures. Benzene is an aromatic hydrocarbon found in crude oil, cigarette smoke and is also used as a solvent in manufacturing industries. Early work by Mele et al., showed association between shoemakers and ET risk, mist likely due to exposure to benzene found in glues used in this profession [89]. Excessive risk for myelofibrosis due to benzene has been suggested in workers in the transport industry [91] with the main source of benzene being petrochemicals. Another study that analyzed several different industries and occupations as a potential source of risk for having an MPN revealed no associations between occupational PAH exposures and risk of MPN [92]. However, this study did find an increased association with extended employment in transportation, communications and public utilities [92].

In addition to benzene, polycyclic aromatic hydrocarbons (PAHs) and particulate matter are contained in emissions produced from the combustion of automobile fuels. Additional studies investigating the association of motor vehicle exhaust with MPN risk would be useful. One of the common sources of benzene, PAHs and particulate matter is cigarette smoke and as described above, smoking has been consistently associated with MPN risk. Future studies in animal models will provide mechanistic insights on the role of cigarette smoke and air pollutants in clonal selection and evolution in MPN disease development. However, indoor sources of chemical exposures including home solvents, paints, bleach, oven cleaner, garden herbicides or pesticides, frequency of boiler maintenance or home car mechanical work were not associated with MPN risk [60]. These differences in findings between occupational and recreational exposures can be attributed to levels and duration of exposures. 

A cluster of 33 PV cases in the Tri-County area of Eastern Pennsylvania caused speculation on possible environmental etiology of MPNs since several hazardous waste exposure sites were identified in this region [93,94]. The incidence of PV in this cluster was four times greater than the rest of Tri-county area [94]. However, an updated and expanded study of PV and other MPNs in the Tri-County area did not identify any PV cancer clusters but an ET cluster of 9 patients was identified [95]. Low case participation affected the study’s strength to find significant disease clustering.

### 4.3. Socioeconomic Factors

The exploratory MOSAICC case-control study was recently used to identify novel factors associated with risk for MPN. MPN patients were found to be more likely to have been raised in an environment where the main household occupation belonged to a lower social class [61]. This is described as childhood socio-economic position (SEP) and lifetime exposures to poor socio-economic conditions can influence adult health outcomes with increased likelihood of chronic conditions such as heart disease [96]. Increased childhood household density was also observed in MPN patients compared to controls [61]. The life course of accumulation of exposure risks attributed to experiencing low socioeconomic conditions such as poor nutrition, infections and exposure to passive smoke can thus predispose to chronic health conditions such as MPN in later life. Lack of educational opportunities owing to socioeconomic status can also influence lifestyle choices, risk behaviors and occupational exposures, further increasing the risk for adverse health outcomes. In fact, the MOSAICC study observed a borderline association between higher education completion and reduced risk for MPN [61]. 

## 5. Lifestyle Modifications as Prevention/Therapy

There is some evidence for risk factors associated with the etiology of MPNs based on retrospective studies. These data suggest that lifestyle modifications to reduce inflammation and/or protect hematopoietic stem cells from stressors may be a useful strategy to prevent progression or development of hematologic malignancies (Figure 2). However, the retrospective studies are far from definitive and large prospective studies are required to establish how specific lifestyle choices impact the development and disease progression of MPN. 

People with early stage MPN may be an ideal target for lifestyle modifications to reduce inflammation. Initial studies should be focused on the primary outcome of symptom reduction to demonstrate the immediate benefits of lifestyle modification for MPN patients. Then, longer large-scale studies could be performed to address how lifestyle modifications shown to be beneficial for symptom reduction impact disease progression and clonal evolution in MPN. 

Manipulation of the environment could steer evolution of hematopoiesis toward a benign outcome. In this regard, the goal is not to necessarily “target” the mutant clone, but to lead hematopoiesis back to polyclonality. This approach may be particularly useful in diseases where directly targeting the mutant clone has not been successful, for example in MPN where JAK inhibitors do not reduce the JAK2 mutant allele burden. Since the *JAK2^V617F^* mutant is prevalent in the general population and is accompanied by higher blood cell counts than mutant negative subjects [35], lifestyle changes such as smoking cessation and high fiber anti-inflammatory diet, will be of considerable importance in this target at risk population.

## 6. Conclusions

Recent findings suggest some role for environmental inflammatory stimuli in the pathogenesis of MPN. Chronic systemic inflammation and oxidative stress induced by these external components may influence disease initiation and clonal expansion or evolution resulting in MPN phenotype. Since an inflammatory environment promotes the selection and expansion of the neoplastic clone, lifestyle modifications in high risk target populations can be recommended as a strategy to halt disease progression or prevent disease development. This is of particular importance in the context of smoking behavior which is positively associated with risk for developing MPN. 

## Figures and Tables

**Figure 1 cancers-12-02038-f001:**
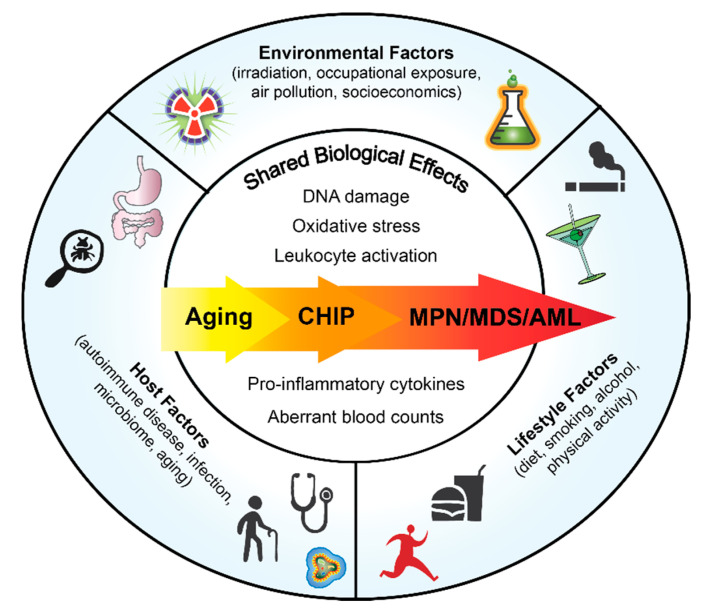
Overview of Factors that Could Impact Normal and Malignant Hematopoiesis.

**Figure 2 cancers-12-02038-f002:**
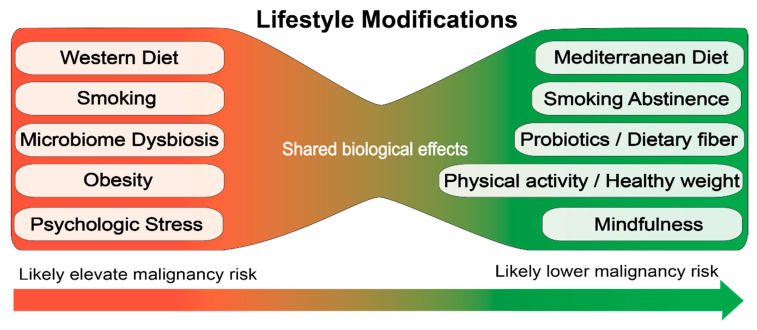
Proposed lifestyle modifications for prevention of hematologic malignancies.

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
