# Peer review of "Impact of Host, Lifestyle and Environmental Factors in the Pathogenesis of MPN"

_cancers, 2020, doi:10.3390/cancers12082038_

Round 1

Reviewer 1 Report

Drs. Ramanathan, Hoover and Fleischman present a review of genetic and non-genetic host factors, lifestyle and environmental factors that influence risk of myeloproliferative neoplasms and their pathogenesis. They provide rationale and a framework for considering some of these modifiable risk factors in the prevention of MPN and other myeloid malignancies. Their work is well organized and comprehensive and this review article should serve as a significant contribution to the field. The following minor suggestions are intended to improve the manuscript.

  1. Abstract - "the existence of sporadic cases" in itself does not indicate that "external factors can play a part in MPN disease development"; this could be explained simply by cell-intrinsic mutation; I agree that external factors play a part but this sentence should be modified.
  2. Lines 27-29 - the description of MPN should indicate that over-produced myeloid cells are mainly mature and/or the authors should distinguish in some other way from MDS and AML, for example.
  3. Lines 33-36 - should be two sentences, not one.
  4. Lines 41-42 and section that follows - CHIP is listed as a "Non-Genetic Host Factor"?? I understand the focus is on CHIP-related inflammation but this is a misleading title. Consider another way to describe.
  5. line 49 - the frequency should be described as "at least" 10%; other studies have shown higher than 10% CHIP prevalence at this age.
  6. Lines 55-57 - some of these studies lack controls and are compared to published control prevalence; some qualification is required.
  7. Lines 99-101 - it would be helpful to have a comparator - for example, 30.5% of the population cohort falls in this age range. 
  8. Line 102 - "prevalence" may be a better term than "rate".
  9. Line 114 - the previous statements describe humans but it should be clear the cited study (36) was performed in mice.
  10. Line 144 and ref. 44 - please check ref. 44 (appears incorrect - about chromosome organization in sperm).
  11. Lines 151-152 - While this may be true, smoking can contribute to mutations. If the authors are to focus on the connection between smoking and chronic inflammation, they should briefly discuss if no evidence has linked mutational signatures associated with tobacco to MPN. One study of CHIP in the oncology setting observed an increased rate of mutations attributed to a smoking signature in oncology treatment-naive CHIP subjects (see PMID: 28803919). On the other hand, a more recent publication found that ASXL1-mutant CHIP was most strongly associated with smoking, but not smoking-related mutational signatures (PMID: 32518416). At least some discussion is warranted.
  12. Lines 189-190 - it might be helpful to discuss this earlier on (see previous comment). Also, this statement contradicts one that follows ("mutagenesis in HSCs can lead to the creation of the JAK2V617F clone" and "the molecular effects [sp. of effects] of smoking on HSCs in the MPN setting have not been evaluated").
  13. Line 202 - If permitted to include non-peer-reviewed/unpublished results, then this should be dampened with "may".
  14. Page 7, first paragraph, and 4.2 Ionizing radiation section - Could the authors speculate about why obesity is associated with ET and not PV? Interesting again about the discordance with radiation exposure and risk of ET vs PV. Is this possible to discuss with the obesity disparity? Is there less advantage of higher JAK2V617 VAF in PV under those environments (and could inflammation be the linkage)?
  15. Sections 5 and 6 - considering how this review may inspire future research directions and possibly clinical trials, can the authors discuss a little more about opportunities for studying these interventions and their potential impact on MPN. For example, could study design be informed by their unpublished studies in MPN dietary intervention? Right now, the review ends rather abruptly and doesn't capitalize on the opportunity to specifically guide the field. Finally, if the authors wish to state this could be "expanded to more common scenarios such as CHIP", then more detail is required. For example, could clonal hematopoiesis be studied in any retrospective cohorts where lifestyle modifications have been implemented, or prospective population cohorts? And what about prevalence of CHIP and expected effect sizes of these interventions, and the sample sizes that would be required for sufficient statistical power? These are just examples. 

Author Response

REVIEWER #1

Drs. Ramanathan, Hoover and Fleischman present a review of genetic and non-genetic host factors, lifestyle and environmental factors that influence risk of myeloproliferative neoplasms and their pathogenesis. They provide rationale and a framework for considering some of these modifiable risk factors in the prevention of MPN and other myeloid malignancies. Their work is well organized and comprehensive and this review article should serve as a significant contribution to the field. The following minor suggestions are intended to improve the manuscript.

  1. Reviewer’s comment:

Abstract - "the existence of sporadic cases" in itself does not indicate that "external factors can play a part in MPN disease development"; this could be explained simply by cell-intrinsic mutation; I agree that external factors play a part but this sentence should be modified.

Authors’ response:

We have removed this sentence, we agree that it isn’t quite correct.

  1. Reviewer’s comment:

Lines 27-29 - the description of MPN should indicate that over-produced myeloid cells are mainly mature and/or the authors should distinguish in some other way from MDS and AML, for example.

Authors’ response:

Lines 27-29 have now been changed to, “Myeloproliferative neoplasms (MPNs) occur as a result of clonal outgrowth of HSCs carrying a somatic mutation most commonly in JAK2 (JAK2V617F), endowing them with cytokine independent growth and ensuing in the overproduction of mature and differentiated cells of the myeloid lineage.”

We have also added lines 29-31, “MPNs are distinct from myelodysplastic syndromes (MDS) and acute myeloid leukemia (AML) which are characterized by arrested differentiation of progenitor cells leading to bone marrow failure and ineffective hematopoiesis.”

  1. Reviewer’s comment:

Lines 33-36 - should be two sentences, not one.

Authors’ response:

Lines 34-37 have been changed to two sentences. “Chronic inflammation is a consequence of both intrinsic and extrinsic factors. Defining how risk factors impact MPN development will enable lifestyle modifications to improve patient outcomes in early stage of disease as well as to prevent MPN initiation in healthy individuals predisposed to developing MPN.”

  1. Reviewer’s comment:

Lines 41-42 and section that follows - CHIP is listed as a "Non-Genetic Host Factor"?? I understand the focus is on CHIP-related inflammation but this is a misleading title. Consider another way to describe.

Authors’ response:

Line 42, “Non-genetic host factors” has been replaced with “Host factors.”

Abstract, lines 16-19 has also been changed to, “Herein we have considered major MPN driver mutation independent host, lifestyle and environmental factors in the pathogenesis of MPN based upon epidemiological and experimental data.”

  1. Reviewer’s comment:

Line 49 - the frequency should be described as "at least" 10%; other studies have shown higher than 10% CHIP prevalence at this age.

Authors’ response:

Line 50 now reads as “at a frequency of at least 10%.”

  1. Reviewer’s comment:

Lines 55-57 - some of these studies lack controls and are compared to published control prevalence; some qualification is required.

Authors’ response:

Lines 56-59, have been revised to, “Compared to the incidence of CHIP in the general population from prior studies and published cohorts, the frequency of CHIP was higher in  chronic inflammatory conditions such as aplastic anemia [15,16], ulcerative colitis patients [17] and in patients with aortic valve stenosis [18].”

  1. Reviewer’s comment:

Lines 99-101 - it would be helpful to have a comparator - for example, 30.5% of the population cohort falls in this age range.

Authors’ response:

Lines 100-102 have been revised to, “The MPN Research Initiative sponsored by 23andMe showed strong age interaction in the epidemiology of MPN where 41.1% of MPN patients were between 61-112 years of age and 31% of were over the age of 60.”

  1. Reviewer’s comment:

Line 102 - "prevalence" may be a better term than "rate".

Authors’ response:

Line 105, the term “rate” has been replaced with “prevalence.”

  1. Reviewer’s comment:

Line 114 - the previous statements describe humans but it should be clear the cited study (36) was performed in mice.

Authors’ response:

Line 116-117 has been changed to, “In mice, the gut microbiome directs innate immune cell development by promoting steady-state myeloid cell development in the bone marrow.”

  1. Reviewer’s comment:

Line 144 and ref. 44 - please check ref. 44 (appears incorrect - about chromosome organization in sperm).

Authors’ response:

Reference #44 has been changed to, “Keshteli AH, Millan B, Madsen KL. Pretreatment with antibiotics may enhance the efficacy of fecal microbiota transplantation in ulcerative colitis: a meta-analysis. Mucosal immunology. 2017;10(2):565-6.”

  1. Reviewer’s comment:

Lines 151-152 - While this may be true, smoking can contribute to mutations. If the authors are to focus on the connection between smoking and chronic inflammation, they should briefly discuss if no evidence has linked mutational signatures associated with tobacco to MPN. One study of CHIP in the oncology setting observed an increased rate of mutations attributed to a smoking signature in oncology treatment-naive CHIP subjects (see PMID: 28803919). On the other hand, a more recent publication found that ASXL1-mutant CHIP was most strongly associated with smoking, but not smoking-related mutational signatures (PMID: 32518416). At least some discussion is warranted.

Authors’ response:

We have now discussed the mutational signature associated with tobacco smoking in the context of MPN following the reviewer’s suggestions. These changes are reflected in lines 153 - 168.  

“A recent report demonstrates that current and heavy smokers in the general population are more likely to carry a JAK2V617F or CALR mutation [35]. Smoking also promotes systemic chronic inflammation via leukocyte activation, endothelial dysfunction, oxidative stress and release of pro-inflammatory cytokines, all characteristic features of MPN [48]. These support the idea that smoking induces mutagenesis in HSCs leading to the creation of a mutant clone and/or produces an environment which allows mutant clones to gain a selective advantage. Coombs et al., observed an increased rate of smoking associated mutational signature in CHIP mutations among smokers with solid tumors (Coombs, 2017, Therapy-related clonal hematopoiesis). Meanwhile, although ASXL1 mutations were predominantly associated with smoking in the UK Biobank cohort, the rate of C>A mutations, characteristic of the smoking related mutational signature 4, was similar between smokers and non-smokers (Dawoud 2020, Clonal myelopoiesis in the UK Biobank). In addition, mutational signature 4 was mainly found in tissues directly exposed to tobacco smoke and could not be extracted from malignancies such as acute myeloid leukemia, despite the known risk associated with smoking (Alexandrov 2016, Mutational signatures associated with tobacco smoking). Thus, it appears that a smoking induced inflammatory environment allows clonal evolution and MPN disease progression.”

  1. Reviewer’s comment:

Lines 189-190 - it might be helpful to discuss this earlier on (see previous comment). Also, this statement contradicts one that follows ("mutagenesis in HSCs can lead to the creation of the JAK2V617F clone" and "the molecular effects [sp. of effects] of smoking on HSCs in the MPN setting have not been evaluated").

Authors’ response:

The changes made following the reviewer’s suggestions above are now on lines 153 - 168.

  1. Reviewer’s comment:

Line 202 - If permitted to include non-peer-reviewed/unpublished results, then this should be dampened with "may".

Authors’ response:

Line 205-206 has been changed to, “These findings indicate that chronic E-cig use may affect hematopoietic stem and progenitor cells and need to be used with caution.”

  1. Reviewer’s comment:

Page 7, first paragraph, and 4.2 Ionizing radiation section - Could the authors speculate about why obesity is associated with ET and not PV? Interesting again about the discordance with radiation exposure and risk of ET vs PV. Is this possible to discuss with the obesity disparity? Is there less advantage of higher JAK2V617 VAF in PV under those environments (and could inflammation be the linkage)?

Authors’ response:

We agree that the disparity in the risk for ET versus PV with obesity and BMI is interesting. Leal et al., speculate that ET and PV possess distinct etiologies with obesity related chronic inflammation playing a mechanistic role in ET (Leal et al., 2013). Obesity is known to affect hematopoiesis (Singer et al., 2014). Interestingly, a very recent publication indicates that increased bone marrow adiposity in mice fed a high fat diet led to enhanced megakaryocyte maturation and accelerated proplatelet formation (Valet et al, Adipocyte fatty acid transfer supports megakaryocyte maturation, Cell reports, July 2020). The influence of adipocyte signaling on platelet generation could be a possible mechanism linking obesity to ET.

This discussion has been included in the main text, lines 278-280, “The disparity between the risk for ET versus PV due to obesity is interesting. One possible mechanism could be due to the influence of increased adiposity in the bone marrow which leads to enhanced megakaryocyte maturation and accelerated proplatelet formation in mice.”

  1. Reviewer’s comment:

Sections 5 and 6 - considering how this review may inspire future research directions and possibly clinical trials, can the authors discuss a little more about opportunities for studying these interventions and their potential impact on MPN. For example, could study design be informed by their unpublished studies in MPN dietary intervention? Right now, the review ends rather abruptly and doesn't capitalize on the opportunity to specifically guide the field. Finally, if the authors wish to state this could be "expanded to more common scenarios such as CHIP", then more detail is required. For example, could clonal hematopoiesis be studied in any retrospective cohorts where lifestyle modifications have been implemented, or prospective population cohorts? And what about prevalence of CHIP and expected effect sizes of these interventions, and the sample sizes that would be required for sufficient statistical power? These are just examples.

Authors’ response:

We have expanded a bit in Section 5 to discuss what sort of trials should be done to evaluate the impact of lifestyle intervention in MPN. We removed the discussion of CHIP, as we think it is best in this article to focus on MPN. Although lifestyle modification in CHIP is definitely of interest to us, to do a discussion justice it may go beyond the scope of this article.

Reviewer 2 Report

This is a well written, concise, adequately referenced review on potential risk factors contributing to the development of MPN. No obvious misinformation is detectable to this reviewer.

Minor issue: Almost all of the data reported is coming from (smaller or larger) retrospective studies. Thus, the data are necessarily largely descriptive. Inherent to the field, almost none of the data is coming from interventional randomized data, and, thus, all recommendations remain "unproven" to some extent. The authors ere, therefore, invited to state generally more cautiously throughout the manuscript the level of certainty of their recommendations and conclusions.

Author Response

REVIEWER #2

This is a well written, concise, adequately referenced review on potential risk factors contributing to the development of MPN. No obvious misinformation is detectable to this reviewer.

  1. Reviewer’s comment:

Minor issue: Almost all of the data reported is coming from (smaller or larger) retrospective studies. Thus, the data are necessarily largely descriptive. Inherent to the field, almost none of the data is coming from interventional randomized data, and, thus, all recommendations remain "unproven" to some extent. The authors ere, therefore, invited to state generally more cautiously throughout the manuscript the level of certainty of their recommendations and conclusions.

Authors’ response:

We have made the following changes to the manuscript to cautiously state our conclusions from retrospective studies.

Line 375, “Recent findings suggest some role for environmental inflammatory stimuli in the pathogenesis of MPN.”

Line 376, “Chronic systemic inflammation and oxidative stress induced by these external components may influence disease initiation and clonal expansion or evolution resulting in MPN phenotype.”